# Innate Immune Status of Glia Modulates Prion Propagation in Early Stage of Infection

**DOI:** 10.3390/cells12141878

**Published:** 2023-07-18

**Authors:** Sang-Gyun Kang, Chiye Kim, Judd Aiken, Debbie McKenzie

**Affiliations:** 1Centre for Prions and Protein Folding Diseases, University of Alberta, Edmonton, AB T6G 2M8, Canada; sanggyun.kang@ualberta.ca (S.-G.K.); chiye@ualberta.ca (C.K.); 2Department of Medicine, University of Alberta, Edmonton, AB T6G 2R7, Canada; 3Department of Biological Sciences, University of Alberta, Edmonton, AB T6G 2E9, Canada; 4Department of Agricultural, Food and Nutritional Sciences, University of Alberta, Edmonton, AB T6G 1C9, Canada

**Keywords:** innate immunity, primary glial cultures, LPS, dexamethasone, hamster, hyper, drowsy

## Abstract

Prion diseases are progressive neurodegenerative disorders affecting humans and various mammals. The prominent neuropathological change in prion-affected brains is neuroinflammation, histopathologically characterized by reactive gliosis surrounding prion deposition. The cause and effect of these cellular responses are still unclear. Here we investigate the impact of innate immune responses on prion replication using in vitro cell culture models. Hamster-adapted transmissible mink encephalopathy prions, hyper (HY) and drowsy (DY) strains, were assayed for accumulation of pathogenic prion protein (PrP^Sc^) in primary glial cultures derived from 8-day-old hamster pups. The kinetics of PrP^Sc^ accumulation largely depended on prion strain and brain regions from where glial cells originated. Glial cells derived from the cerebellum were susceptible to HY, but resistant to DY strain as determined by western blot analysis, immunocytochemistry, and animal bioassay. Glial cells from the cerebral cortex were, however, refractory to both strains. PrP^Sc^ accumulation was affected by innate immune modulators. Priming glial cells with lipopolysaccharide decreased prion replication, whereas pre-treatment with dexamethasone, inhibiting innate immunity, increased susceptibility to DY infection. Our results suggest that neuroinflammation resulting from prion infection is a response to resolve and/or prevent prion propagation in the brain. It implies a therapeutic potential of innate immune modulation in the early stages of prion disease.

## 1. Introduction

Prion diseases are fatal neurodegenerative disorders characterized by the accumulation of abnormal isoforms of prion protein (PrP^Sc^), predominantly in the brain [1]. The precise etiology of prion diseases is not fully understood, but involves the conversion of the normal, cellular form of the prion protein (PrP^C^) into the abnormal, infectious forms, PrP^Sc^. This conversion can occur spontaneously or be triggered by exposure to PrP^Sc^, either through infection or by inherited genetic mutations that increase the risk of developing prion diseases [2,3]. The increase in pathogenic prion isoforms resulting from the continuous conversion of PrP^C^ into PrP^Sc^ harms neurons and causes chronic inflammation [4,5]. Neuroinflammation is a process that involves the activation of immune cells and the release of inflammatory mediators in response to various pathological stimuli, including protein aggregation [6]. In neurodegenerative diseases, such as Alzheimer’s disease, tauopathies, Parkinson’s disease, Huntington’s disease, and prion diseases, the accumulation of abnormally folded proteins triggers immune responses, inducing microgliosis and astrogliosis [7,8,9,10,11,12,13,14]. Persistent activation of these glial cells and resulting chronic inflammation can lead to structural and functional damage to neurons, contributing to the development of neurodegenerative diseases [15].

The reactive gliosis—activation of microglia and astrocytes—triggered by pathogenic prion conformers can have both positive and negative consequences, depending on the severity and timing of the inflammatory response. Early and short-lived local neuroinflammation can be beneficial. Microglia can have a protective role against prion infection, as observed in experiments involving cerebellar organotypic cultures and transgenic animals. Pharmacogenetic depletion of microglia increases neurotoxicity, shortening the incubation time of RML-infected mice [16]. The inhibition of the colony stimulating factor 1 receptor eliminates microglia in prion-infected brains, shortening disease duration and increasing vacuolation, astrogliosis, and PrP^Sc^ deposition [17]. The triple-knockout of microglial proinflammatory effector molecules accelerates the progression of prion disease in transgenic animals [18], indicating the protective role of microglia.

The detrimental effects of glial response can be more severe and prolonged, particularly during the later stages of the disease. Gene expression profiling of various brain regions of prion-infected animals shows a strong inverse correlation between the incubation time and the activation status of astrocytes, affecting their physiological functions [19]. Reactive astrocytes and microglia from prion-infected mice hinder the development of dendritic spines and the maturation of synapses in cultured neurons [14]. Moreover, glial cells in mice with terminal prion disease showed extensive changes in translational profiles, while neurons displayed only minimal changes [20]. These data indicate that glial cells may be the primary drivers of neurological damage associated with prion diseases. In normal conditions, the primary immune response is triggered in response to various pathogens and endogenous danger signals that are localized to the site of damage and resolved quickly, making it challenging to evaluate their impact on disease progression. However, the initial acute inflammatory response to pathogenic protein folding in various neurodegenerative diseases is considered a defensive mechanism that aids in the elimination of these proteins and the restoration of tissue homeostasis [21].

PrP^Sc^ is present in brain cells such as neurons, microglia, and astrocytes [22], and its cellular tropism is influenced by various factors such as prion strains, affected brain regions, and stages of the disease [23]. Astrocytes have garnered significant research attention as they are relatively easy to handle and support prion propagation in culture (as reviewed in [24]). Interestingly, the accumulation of PrP^Sc^ in astrocytes occurs before the onset of neuropathological changes in prion-affected brains [23,25]. These findings suggest that astrocytes, the most abundant and largest type of glial cells in the CNS, may play a crucial role in the initiation and progression of prion diseases. Nonetheless, the precise mechanisms underlying their involvement in disease pathogenesis are still not fully understood.

Our previous research has focused on the role of glial cells in regulating prion replication during the early stages of infection. We demonstrated that activation of the innate immune response through toll-like receptor (TLR) signaling could restrict prion propagation in cultured glial cells [26]. Expanding upon this work, we aimed to investigate the replication of two distinct prion strains—hyper (HY) and drowsy (DY)—in primary glial cultures derived from hamsters. This study revealed that the innate immune status of glial cells determines whether prions are cleared or allowed to self-propagate, suggesting that targeting the innate immune system via glial cell modulation could have therapeutic potential in the treatment of prion diseases.

## 2. Materials and Methods

### 2.1. The Primary Cell Cultures and Prion Exposure

Primary cerebellar granule neurons and glial cells were extracted from 7-day-old wild-type Syrian golden hamsters, FVB/N mice, or Tg7 transgenic mice expressing hamster PrP^C^, by mechanical and enzymatic dissociation of the cerebella. The cells were then plated on plastic tissue culture vessels coated with 10 μg/mL Poly-L-lysine (PLL, Sigma, MO, USA, P4747) at a density of 2 × 105 cells/cm^2^ and cultured in MEM (Sigma, M4655) containing 10% fetal bovine serum (FBS, Gibco, 12438), 25 mM KCl, penicillin and streptomycin (Gibco, MT, USA, 15140). The medium was completely replaced after 24 h and subsequently changed twice a week.

Primary hamster glial cultures were established by culturing the mixture of granule neurons and glial cells described above for 4–6 weeks. Once the cells reached confluence, the cells were trypsinized and replated on PLL-coated culture vessels. After an additional 2 weeks of in vitro culture, the cells were cryopreserved until use. The expression of glial cell markers was confirmed by immunocytochemistry and western blot analyses for glial fibrillary acidic protein (GFAP, abcam, MN, USA, ab4674), glutamine synthetase (GS, Millipore, MA, USA, MAB302), cluster of differentiation molecule 11b (CD11b, abcam, ab133357), and neuron-specific nuclear protein (NeuN, Millipore, MAB377). Primary glial cultures derived from cerebral cortex were established from 1- to 4-day-old Syrian golden hamsters using the same protocol.

The propagation of prions was investigated using the primary cultures. The cultures were optionally treated with an innate immune stimulator or suppressor for 24 h, and then washed twice with PBS. Next, the cultures were exposed to prion-affected brain homogenates at final concentrations of 2–20 μg/mL total protein in the culture medium. Following exposure for 24 h, the cultures were washed twice with PBS to remove residual prions and maintained in fresh culture media. At 28 to 35 days post exposure, the cells were harvested and lysed in RIPA lysis buffer containing Protease Inhibitor Cocktail. The RIPA lysis buffer included 1% Triton X-100, 1% sodium deoxycholate, 150 mM NaCl, 50 mM Tris-HCl (pH 7.4), 0.1% SDS, and 1 mM EDTA.

### 2.2. Prion Preparation (Prion-Affected Brain Homogenates)

Syrian golden hamsters, intracerebrally inoculated with hyper (HY) and drowsy (DY) strains of hamster-adapted transmissible mink encephalopathy (TME), were euthanized at the clinical stage of disease. Hamster brains were homogenized in PBS and centrifuged at 1000 × *g* for 5 min to eliminate non-homogenized debris. The BCA protein assay kit (Pierce, WI, USA, 23235) was utilized to measure the total protein concentration of the clarified brain homogenate, which was then adjusted to a final concentration of 2 mg/mL. Brain homogenates were aliquoted and then stored at −80 °C until use. In addition to the TME strains, the study also included mouse-adapted scrapie strains, RML, 22L, and ME7, and hamster- and mouse-adapted cervid prion strains, Wisc1, 95H+, and CWD-Elk. Non-infectious brain homogenates (NBHs) served as negative controls.

### 2.3. Western Blot

Protein quantification was performed using BCA protein assay (Pierce, 23235) on cell lysates and brain homogenates. For the detection of the abnormal isoform of prion protein (PrP^Sc^), samples were treated with proteinase K (PK, Sigma, P4850) at 50 μg/mL for 30 min at 37 °C, unless otherwise indicated. The samples were resolved on 12% acrylamide precast gels (Invitrogen, MA, USA, NP0342) and transferred to PVDF membrane. The membrane was blocked using 5% non-fat dry milk in TBST (TBS with 0.1% Tween 20) and probed overnight at 4 °C using primary antibodies: 3F10 anti-PrP monoclonal antibody [27], 3F4 anti-PrP monoclonal antibody (a kind gift from Richard Rubenstein), Bar224 anti-PrP monoclonal antibody (bertin Bioreagent, MD, USA), anti-GFAP polyclonal antibody, anti-GS monoclonal antibody, anti-CD11b monoclonal antibody, anti- NeuN monoclonal antibody, and anti-β-actin monoclonal antibody (abcam, ab20272). The secondary antibodies used were anti-mouse, rabbit, or chicken immunoglobulin conjugated to horseradish peroxidase (Bio-Rad, CA, USA, 170-6516; Bio-Rad, 170-6515; Abcam, ab97135) or alkaline phosphatase (Promega, WI, USA, S327B or S323B). The target proteins were detected by chemiluminescent (Pierce, 32209) or fluorescent (Promega, S1000) signals. The membranes were then stripped using a western blot stripping buffer (Thermo Scientific, Waltham, MA, USA, 46430) and re-probed as necessary.

### 2.4. Immunocytochemistry

Immunocytochemistry was performed on cells plated on PLL-coated glass chamber slides (Nalge Nunc, NY, USA, 154526). To prepare the cells for immunostaining, they were fixed with paraformaldehyde (4%, pH 7.4) for 10 min and permeabilized with PBS containing TritonX-100 (0.1%). To expose epitopes of PrP^Sc^, cells were treated with guanidine thiocyanate (GdnSCN, 3M) for 2 h at 4 °C. The fixed cells were then blocked with 1% bovine serum albumin (BSA) in PBST (PBS with 0.1% Tween 20) for 30 min. Immunostaining was carried out using SAF83 anti-PrP monoclonal antibody (Cayman, MI, USA, 189765) to probe for PrP^Sc^ and anti-GFAP polyclonal antibody (Abcam, ab4674) to label astrocytes. Appropriate fluorescent-conjugated secondary antibodies (Alexa Fluor 488 or Alexa Fluor 594, Invitrogen) were used to visualize the target molecules. Nuclear counterstain was performed with 4′,6-diamidino-2-phenylindole (DAPI) fluorescent stain (Invitrogen, P36935).

### 2.5. Cell Viability Assay

Cell viability was assessed by measuring lactate dehydrogenase (LDH) activity in the culture supernatant using a commercial kit (Promega, G1780) following the manufacturer’s instructions. After incubating the culture supernatant with a tetrazolium salt substrate for 30 min at room temperature, the enzymatic reaction’s red formazan products were measured using a microtiter plate reader (μQuant, Bio-Tek, VT, USA) at a wavelength of 490 nm. The LDH activity was quantified and expressed as a percentage relative to control samples.

### 2.6. Animal Bioassay

All animal handling and maintenance were conducted under the Biosafety Level 2 Enhanced (BSL2+) facility at the Centre for Prions and Protein Folding Diseases in accordance with the University of Alberta Animal Care and Use Committee’s approved animal use protocols (AUP00000914). The inocula were diluted in PBS, and 25 μL of each preparation was intracranially administered to male Syrian golden hamsters. The animals were monitored daily and were euthanized when clinical disease was established.

### 2.7. Statistical Analysis

The number of biological and technical replicates used for each observation in compared groups was indicated in the corresponding figure legends. The sample size (*n*) represents biological replicates and ranged from 2 to 8. Statistical analysis was conducted using PRISM version 5 software (GraphPad Software, CA, USA). For most comparisons of means, the unpaired, two-tailed Student *t* test was employed. The log-rank test and ANOVA with post hoc Tukey’s multiple comparison test was used to analyze the data from the animal bioassay (Table 1 and Figure 12).

## 3. Results

### 3.1. Differential Susceptibility of Primary Glial Cells to Prion Strains

Primary glial cultures were isolated from 8-day-old hamster pups and cryopreserved at confluence. After 7 days of in vitro culture, the cells were harvested (Figure 1a). Analysis of cellular marker expression, in the primary glial cultures, by western blot (Figure 1b) showed higher levels of glial fibrillary acidic protein (GFAP) and a cluster of differentiation molecule 11b (CD11b), but lower levels of glutamine synthetase (GS) compared to hamster brain homogenates. Neuron-specific nuclear protein (NeuN) expression was not detectable, indicating the absence of neuronal cells in the cultures. We confirmed the specificity of the antibodies by comparing the primary glial cells from mice and mouse brain homogenates. The analysis suggests that the primary glial cultures consist primarily of astrocytes and microglial cells, without neuronal cells.

The accumulation of PrP^Sc^ in the cerebellum of hamsters infected with HY and DY strains [28,29] indicates that these strains could infect cerebellar glial cells in in vitro cultures. The exposure of primary glial cultures to HY and DY prions resulted in the successful establishment of HY infection but not DY infection (Figure 2). Western blot analyses of PrP^Sc^ in cell lysates, following proteinase K (PK) digestion (at 50 μg/mL for 30 min at 37 °C), confirmed HY infection and replication in the glial cells (Figure 2a,b). The lack of PrP^Sc^ accumulation in the DY-exposed glial culture, even at 14 weeks post exposure (Figure 2c,d) suggests that DY strain may have a lower affinity for glial cells or may not be able to replicate efficiently in these cells. These strain-dependent susceptibilities of glia in cultures may provide valuable insights into the potential role of glial cells in prion replication and stable infection.

Since DY prions are less resistant to PK digestion than HY prions [30], it is possible that DY was in the glial cultures, but the PK treatment abolished DY signals. To address this possibility, resistance of DY prion to PK digestion was examined. The results in Figure 3 showed that 100 µg/mL of PK reduced PrP^Sc^ signals in DY-infected hamster brain homogenates (the inoculum), but detectable signals were still present (Figure 3a,b). However, when the DY-infected cell lysates were treated with various concentrations of PK, PrP^Sc^ was undetectable even at the lowest concentration of PK (25 µg/mL) (Figure 3c), confirming that the cultured glial cells are refractory to DY infection.

Cellular tropism is a well-established factor in prion biology [31,32]. To investigate the replication of HY and DY prions in the presence of neuronal cells, we tested primary mixed cultures of granule neurons and glial cells (GN-Glia). Similar to the glial culture results (Figure 1), GN-Glia were susceptible to infection by HY but not DY prions (Figure 4a). Interestingly, primary glial cells isolated from the cerebral cortex were resistant to both HY and DY strains (Figure 4b), highlighting the importance of cellular tropism in prion diseases. GN-Gila cultures obtained from a transgenic line Tg7 expressing the cellular isoform of hamster prion protein (PrP^C^) instead of endogenous mouse PrP^C^ showed a similar pattern of HY prion accumulation (Figure 4c). Notably, early time points showed the presence of DY signals (Figure 4d), suggesting that the DY prion was taken up by glial cells, but was either not able to replicate successfully or was cleared from the cells. HY and DY prions were propagated differently in the primary cultures, with HY prion showing faster and more efficient propagation in the cell cultures tested. The observation that DY signals were detected at early time points but not at later time points raises questions about the dynamics of prion replication and clearance in glial cells.

We next investigated the strain-dependent susceptibility of primary glial cultures. The hamster glial cells were exposed to various prion isolates, including mouse- and hamster-adapted deer prions and mouse-adapted scrapie prions (Figure 5a,b). Hamster-adapted deer prion strain, Wisc1, was capable of replicating in the hamster glial cells, but mouse (Tg33)-adapted Wisc1 could not (Figure 5c). This result emphasizes the importance of the host prion protein (PrP^C^) in establishing prion infection. None of the mouse prion strains tested replicated in the hamster glial cultures (Figure 5d), indicating that the glial cells recapitulate species barriers seen in animal models.

### 3.2. Effects of Prion Propagation on Glial Marker Expression

Susceptibility of the glial cells to HY prion was confirmed using immunocytochemistry. HY prion accumulation appeared as strong puncta staining located near the nucleus in ramified glial cells (Figure 6). It was notable that both GFAP-expressing glial cells (middle panels in Figure 6) and GFAP-negative cells exhibited PrP^Sc^ accumulation (top panels in Figure 7). Moreover, we identified PrP^Sc^-positive elongated membrane projections located between two distinct cells (bottom panels in Figure 7), which resembled tunneling nanotubes previously implicated in the propagation of pathogenic PrP^Sc^ throughout the nervous system [33].

The effect of PrP^Sc^ accumulation on cultured glial populations was assessed by analyzing the expression levels of cellular markers using western blot analysis (Figure 8a,b). No significant differences in the levels of GFAP and CD11b were identified, suggesting that astrocytes and microglia were not affected by PrP^Sc^ accumulation. However, the expression of GS was found to be decreased in conjunction with HY infection, indicating that there may be an impact on glutamate metabolism. The levels of LRP1, a receptor involved in the binding and internalization of oligomeric forms of PrP^Sc^, α-synuclein, and pathogenic tau [34,35,36], remained consistent regardless of PrP^Sc^ replication. It implies that the failure in DY propagation in this glial culture system was not likely due to the alteration and/or inhibition of the internalization process of DY prions.

### 3.3. Innate Immune Status of Glial Cells Determines Prion Propagation

To further investigate the impact of glial immune status on PrP^Sc^ accumulation, glial cells were stimulated with toll-like receptor (TLR) ligands, lipopolysaccharides (LPS, 10 µg/mL) or polyinosinic:polycytidylic acid (poly I:C, 100 µg/mL) (Figure 8c). These ligands trigger myeloid differentiation primary response protein 88 (MyD88)-dependent and -independent toll-like receptors (TLRs) signaling, respectively. After priming the glial cell cultures with LPS for 24 h, there was a significant decrease in the levels of PrP^Sc^ compared to mock-treated controls following 4 weeks of incubation with HY prion (Figure 9a). Conversely, pre-treatment with poly I:C had no impact on PrP^Sc^ levels. In addition to affecting PrP^Sc^ accumulation, LPS-stimulation also decreased the expression of astrocytic markers (GFAP and GS), the microglial marker CD11b, and LRP1 (Figure 9b,c). These findings suggest that LPS-priming may have a protective effect against HY PrP^Sc^ accumulation.

Given that innate immune stimulation appears to protect against HY prion infection in glial cells, we hypothesized that innate immune suppression might allow DY prion replication. To test this, glial cells were pre-treated with dexamethasone (Dex), a synthetic corticosteroid that effectively suppresses LPS-mediated innate immune response in primary glial cells derived from mouse cerebella (GLIA 2016) (Figure 10a). Dex treatment significantly increased the levels of PrP^Sc^, compared to mock-treated controls, following 4 weeks of incubation with DY prion (Figure 10b,c), indicating that suppression of innate immunity may promote or allow DY prion replication. Dex treatment also altered the expression of glial markers, with a decrease in GFAP and CD11b and an increase in GS. However, there was no difference observed in the level of LRP1 (Figure 10b,c).

These observations led us to speculate that although the DY prion was unable to replicate in glial cultures, exposure to DY might influence HY prion infection. To investigate this, glial cells were exposed to DY prions for 24 h, then exposed to HY prions for four weeks (as illustrated in Figure 11a). The pre-exposure to DY prions caused a slight but significant reduction in HY prion infection after 4 weeks of incubation (Figure 11b,c). These findings suggest that while the DY prion was not able to establish an infection in the glial cells, it did interfere with the HY prion infection.

### 3.4. PrP^Sc^ Produced in Glial Cells Correlated with the Prion Infectivity

Prion infectivity of glial cells infected with HY and DY strains were determined by animal bioassay (Table 1, Figure 12a,b). HY-infected glial cell lysates (at 100 µg/mL concentration) induced disease onset with an average incubation period of 110 ± 5 days (n = 8), which was similar to HY-infected brain homogenate (at 10 µg/mL concentration) with incubation times of 106 ± 2 days (n = 8). Reduction in PrP^Sc^ in HY-infected glial cells following LPS-priming significantly increased the incubation period (117 ± 7 days, n = 8) compared to HY-infected glial cells (Figure 12a,c). In contrast, DY-infected glial cells following immune suppression with Dex (at 100 µg/mL concentration) resulted in clinical prion disease with an average incubation period of 261 ± 7 days (n = 4) (Figure 12b). However, glial cells exposed to DY prion did not develop disease until the end of the experiment (365 days). Notably, one of the three animals inoculated with DY-exposed glial lysates exhibited PrP^Sc^ signals in post-mortem analysis of brains (Figure 12d), indicating a potential infectivity close to the median lethal dose (LD50). None of the animals inoculated with uninfected brain homogenate or control glial lysates caused disease (within 365 days). The results indicate that primary glial cultures infected to HY and DY prions were able to transmit disease to the hamsters and that the amount of PrP^Sc^ produced by the glial cells correlates with the prion infectivity.

**Table 1 cells-12-01878-t001:** Animal bioassay for HY and DY accumulated in immune modulated primary glial cultures.

Inoculum	BH HY	BH HY	BH DY	BH -	Glia LPS-HY	Glia I:C-HY	Glia HY	Glia -	Glia Dex-DY	Glia DY	PBS -
Conc. (µg/µL)	1	0.01	1	1	0.1	0.1	0.1	0.1	0.1	0.1	-
Attack rate	4/4	8/8	8/8	0/4	8/8	8/8	8/8	0/4	4/4	1/3	0/3
Latency (days)	86	102	204	-	108	106	102	-	245	365	-
	91	104	210		108	106	104		255		
	91	106	219		112	107	107		257		
	91	106	224		121	108	112		285		
		107	224		122	112	113				
		107	233		122	113	113				
		107	245		123	113	115				
		108	281		123	115	116				
Mean ± SD	90 ± 3	106 ± 2	230 ± 24	>365	117 ± 7 *	110 ± 4	110 ± 5	>365	261 ± 17 ***	>365	>365

Male Syrian golden hamsters were intracerebrally inoculated with brain homogenates or glial lysates at the indicated concentrations of total protein. BH, brain homogenate; Glia, glial cell lysates; LPS-HY and I:C-HY, primed glial cells with LPS and poly I:C prior to infection with HY prion, respectively; Dex-DY, suppressed glial cells with dexamethasone prior to infection with DY prion. * *p* < 0.05 and *** *p* < 0.001 in comparison with Glia-HY or Glia-DY controls, respectively.

We analyzed cellular markers using western blotting to investigate the glial populations in hamster brains affected by prion infection (Figure 13). Our results indicated that the accumulation of PrP^Sc^ was inversely related to the expression of NeuN but was directly related to the expression of GFAP and CD11b. Interestingly, in contrast to what was observed in primary glial cultures, the levels of GFAP and CD11b in hamster brains were influenced by the accumulation of PrP^Sc^, resulting in increased levels of GFAP and CD11b. These findings suggest that glial cells play a role in prion pathogenesis, and that the interplay between neurons, astrocytes, and microglia could be more complex in vivo compared to in vitro cultured cells.

## 4. Discussion

Most research on microglia and astrocytes in the field of prion biology has focused on their role in disease progression, including their phenotypic changes from physiological to neurotoxic reactive gliosis, and their effect on incubation periods [13,14,16,17,18,20,23,37,38,39,40]. In this study, we investigated the innate immune defense against pathogenic prions in the early stages when the nucleation and replication of prions occur.

Glial cells derived from the cerebella of hamsters were found to have differential susceptibility to HY and DY prion strains, with high susceptibility to HY and resistance to DY. Glial cells from the cerebral cortex, however, were refractory to both strains. This selective targeting of certain cell types and brain regions is a defining characteristic of prion strains. To investigate whether species-specific effects could account for the strain-dependent susceptibility observed, primary mixed cultures of granule neurons and glial cells from hamsters and Tg7 transgenic mice (expressing hamster PrP^C^) were compared. The same pattern of susceptibility was observed in both cultures, indicating that the results are reliable and that unidentified cellular machinery, in addition to PrP^C^, plays an active role in determining susceptibility to individual prion strains. Interestingly, strong DY signals, roughly equivalent to HY signals, were detected in neuron–glia mixed cultures at 7 days post-infection, but these DY signals decreased over time. This decrease in DY could be due to a decline in the number of granule neurons in the cultures [26], implying a possible neuron-specific tropism of DY. Further studies are required to validate this observation.

In late-stage prion diseases, astrocytes and microglia enter a reactive state often characterized by increased expression of GFAP and Iba1/CD11b, respectively [41]. However, our study on primary glia cultures found no effect of HY and DY prions on the expression of GFAP and GS in astrocytes nor CD11b in microglia. This suggests that PrP^Sc^ accumulation alone may not be the sole factor triggering GFAP and CD11b expression in the diseased brains. It is possible that a complex interplay between different types of brain cells contributes to the expression of GFAP and CD11b [42]. Studying the cellular crosstalk involved in the altering expression of astrocytic and microglial markers using glial cells directly isolated from hamsters presents several challenges. The lack of validated antibodies for hamster proteins can make it difficult to accurately measure specific targets using immunoassays. Despite these challenges, primary glial cultures derived from hamsters remain a valuable tool for investigating the behavior of HY and DY prion strains, without requiring genetic modifications of cells or animals, which may lead to unintended off target effects.

Mainly, studies have shown that the replication of HY prions occurs in the lymphoreticular system (LRS) before invading the nervous system during peripheral infection, while DY replications have not been detected in the LRS regardless of the routes of infection [32,43,44,45]. Unknown biological processes or cellular components in living cells have been proposed to explain why HY and DY prion strains exhibit different tropisms for lymphoid tissues [29,46]. Our data suggest that DY replication was blocked by innate immune responses in the lymphoid tissues whereas HY was able to replicate in the same immune milieu. In brains with neurodegenerative conditions, astrocytes, along with microglia, are immune effector cells that respond to harmful substances such as abnormally folded pathogenic proteins [47]. We demonstrate that the kinetics of PrP^Sc^ accumulation could be altered by innate immune status of glia as a determinant of initial prion infection. When glial cells were primed with LPS, prion replication decreased, whereas pre-treatment with dexamethasone to inhibit innate immunity increased susceptibility to DY infection (Figure 14).

DY prion infection impedes HY superinfection when administered through peripheral nerves [46,48]. This led us to propose that DY prions could activate glial immune responses, and may hinder HY infection, even if DY are not capable of propagating in glial cells. To test this hypothesis, we exposed glial cultures to a 10-fold higher concentration of DY prions before infecting them with HY. We observed a significant decrease in HY PrP^Sc^ accumulation after 24 h of incubation with DY; however, the ability to block HY replication was not as robust as previously reported in in vivo models. This could be due to insufficient incubation time with DY, undetectable levels of DY PrP^Sc^ in the pre-incubated glial cells, or the need for a higher ratio of DY to HY prions to achieve effective inhibition, as shown in previous studies [46,48]. While our attempt to understand the mechanism of prion strain interference observed in vivo is promising, further experiments are necessary to confirm this intriguing phenomenon.

Astrocytes are the first glial cells to accumulate PrP^Sc^ deposits in prion-infected brains, prior to the onset of neuropathological changes [23,25]. Additionally, glial cells exhibit significant changes in their gene expression profile as the disease progresses [20]. Our findings suggest that the immune response of glial cells plays a crucial role in the initiation of prion replication, especially in the early stages of the disease, and is therefore critical in the prion pathogenesis. We propose that the neuroinflammation observed in prion-diseased brains is an attempt by the immune system to resolve and/or prevent the spread of prions in the brain. Our results highlight the potential for therapeutic interventions targeting the innate immune response in the early stages of prion disease.

## 5. Conclusions

Our study highlights the significant role of glial cells in prion replication. We found differential susceptibility of glial cells to different prion strains, manifested by the innate immune status of the glial cells. Glial immune conditions determine prion propagation during the early stage of infection. Moreover, the data imply that neuroinflammation found in prion-affected brains may represent a defensive response against pathogenic prion proteins, providing protection in the early stages, but becoming detrimental in chronic stages. These findings suggest the potential for therapeutic interventions aimed at modulating the innate immune system during the early stages of prion disease.

## Figures and Tables

**Figure 1 cells-12-01878-f001:**
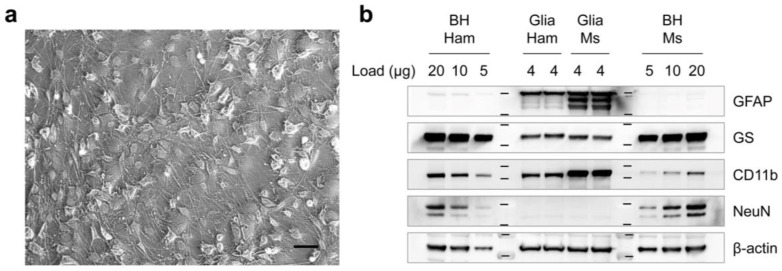
Primary glial cultures derived from hamsters. (**a**) A micrograph of primary glial cells obtained from cerebella of 8-day-old hamster pups that were cryopreserved and then cultured in vitro for 7 days. Scale bar, 50 µm. (**b**) Characterization of the glial cell population. Hamster primary glial cells from (**a**) were analyzed for the expression of astrocytic and microglial markers, including glial fibrillary acidic protein (GFAP), glutamine synthetase (GS), cluster of differentiation molecule 11b (CD11b), and neuron-specific nuclear protein (NeuN), after 7 days of in vitro culture. Primary glial cells derived from mice were used as positive controls for the antibodies tested. Brain homogenates (BHs) were included as references. β-actin, a loading control; Ham, hamster; Ms, mouse.

**Figure 2 cells-12-01878-f002:**
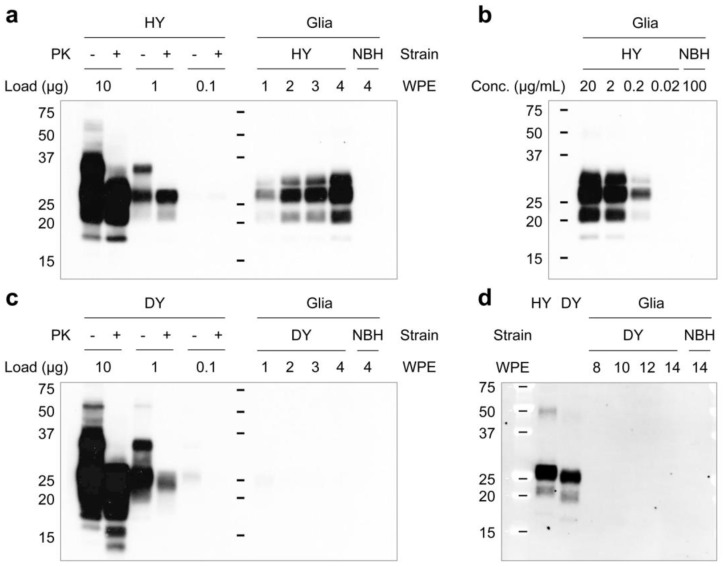
Kinetics of prion propagation in glial cultures. The kinetics of prion propagation in primary glial cultures were determined by western blot analyses using 3F10 anti-PrP monoclonal antibody, with proteinase (PK) digestion as an optional step. (**a**) Primary glial cells, exposed to brain homogenate derived from terminally ill HY prion-infected hamsters (HY) at a final concentration of 2 µg/mL, were harvested at the indicated time points for analysis. (**b**) Dose-dependent HY prion propagation was observed. The primary glial cultures were exposed to HY prion at the indicated concentrations and harvested at 4 weeks post exposure (WPE). (**c**,**d**) The kinetics of DY prion propagation in primary glial cultures. Cells were exposed to DY prion-affected hamster brain homogenate (DY) at a final concentration of 20 µg/mL and harvested at the indicated time points. Non-infectious brain homogenate (NBH) was included as a negative control. Molecular weights are presented in kilodaltons (kDa).

**Figure 3 cells-12-01878-f003:**
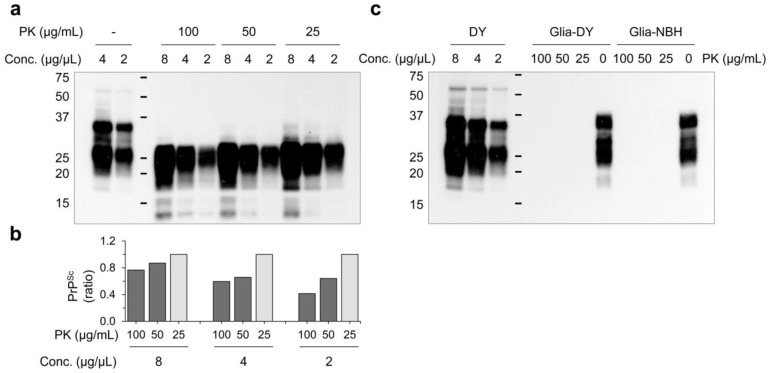
Resistance of DY prion to proteinase K (PK) digestion. (**a**) Different dilutions of DY-infected hamster brain homogenates (DY) were treated with various concentrations of PK, and the presence of PrP^Sc^ was monitored by western blot using 3F10 anti-PrP monoclonal antibody. (**b**) The intensity of the western blot results from (**a**) was quantified and expressed as ratios relative to the intensity obtained with PK at 25 µg/mL. (**c**) Primary glial cells were exposed to DY prion and harvested at 4 weeks post exposure. The cell lysates were treated with PK at the same concentrations used in (**a**), and PrP^Sc^ was analyzed by western blot using 3F10 anti-PrP monoclonal antibody. Non-infectious brain homogenate (NBH) was used as a negative control.

**Figure 4 cells-12-01878-f004:**
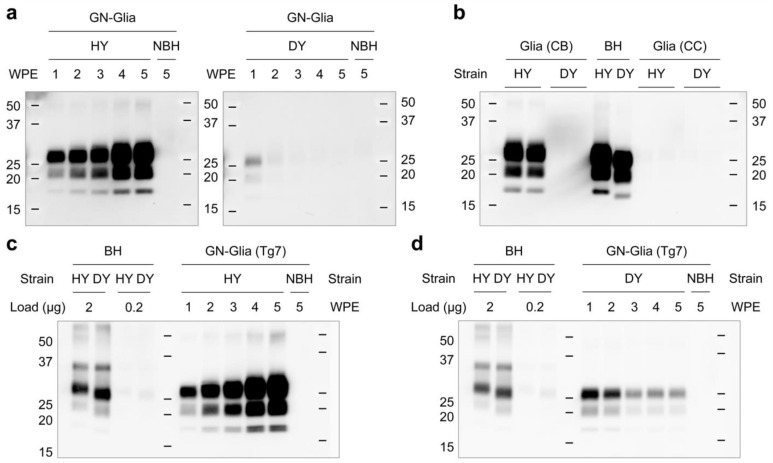
Kinetics of prion propagation in mixed neuronal and glial cultures. Primary mixed neuronal and glial cultures were exposed to HY (2 µg/mL) and DY prion (20 µg/mL) following 7 days’ in vitro culture. The cells were harvested at the indicated time points and PrP^Sc^ was analyzed by western blot with 3F10 anti-PrP monoclonal antibody after PK digestion. (**a**) The primary cerebellar granule neurons and glial cells (GN-Glia) were derived from 8-day-old hamsters and exposed to HY and DY prions. The cells were harvested at indicated time points for analysis. (**b**) The primary glial cells derived from cerebral cortexes (CC) were exposed to HY and DY prion and harvested at 4 weeks post exposure. CB, cerebellum. (**c**,**d**) The primary GN-Glia cultures derived from Tg7 transgenic mice that express hamster cellular isoform of prion protein (PrP^C^). The cells were exposed to HY (**c**) and DY (**d**) prions and harvested at indicated time points for analysis. HY and DY brain homogenates were included as references. NBH, non-infectious brain homogenate.

**Figure 5 cells-12-01878-f005:**
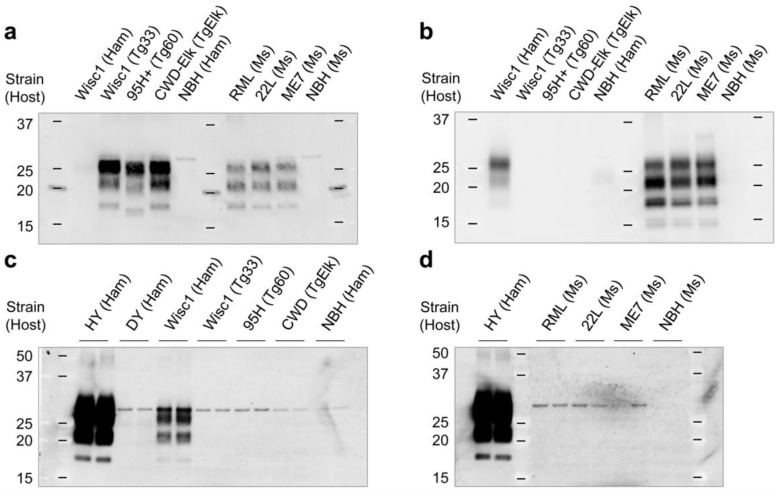
Susceptibility of glial cultures to various prion strains. (**a**,**b**) Western blot analyses of prion-affected hamster and mouse brain homogenates infected with different prion strains. The brain homogenates were treated with proteinase K (PK) and then analyzed by western blot using anti-PrP monoclonal antibodies, Bar224 (**a**) and 3F10 (**b**), to detect PrP^Sc^. The different prion strains used in this study included hamster- and mouse-adapted cervid prion strains, Wisc1, 95H+, and CWD-Elk, and mouse-adapted scrapie strains, RML, 22L, and ME7. Non-infectious brain homogenates (NBHs) were used as negative controls. (**c**,**d**) PrP^Sc^ accumulation in primary glial cultures exposed to prion-affected brain homogenates shown in (**a**,**b**). The cells were harvested 4 weeks post exposure and analyzed by western blot with 3F10 anti-PrP monoclonal antibody to detect PrP^Sc^ accumulation. Tg33 and Tg60, transgenic mice expressing white-tailed deer PrP^C^ with G96 (wild type) and S96 polymorphisms, respectively; TgElk, a transgenic mouse line expressing elk PrP^C^.

**Figure 6 cells-12-01878-f006:**
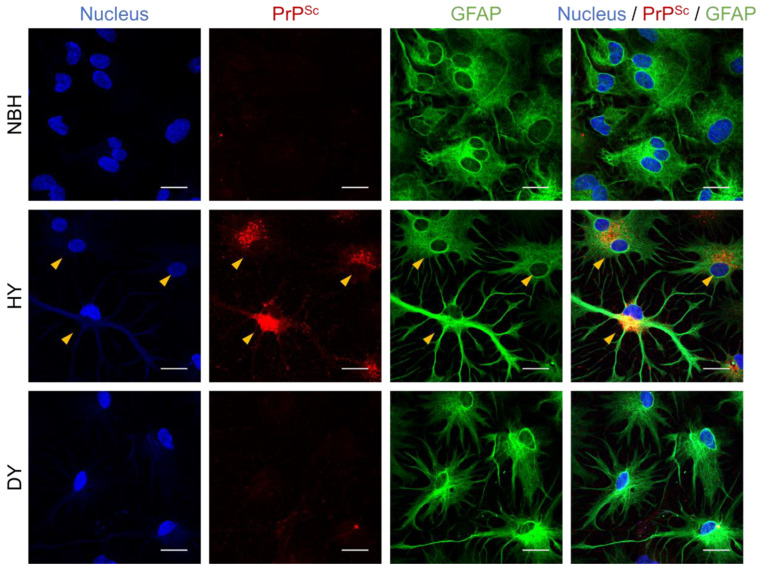
Immunocytochemistry of PrPSc in primary glial cultures exposed to different prion strains. Primary glial cultures were exposed to non-infectious brain homogenate (NBH), HY, and DY prions. After 4 weeks of exposure, the cells were fixed and subjected to immunocytochemistry using SAF83 anti-PrP monoclonal antibody to detect PrP^Sc^. PrP^Sc^ appeared as intensive punctate signals in red (yellow arrowheads), while glial fibrillary acidic protein (GFAP) was used as a marker for astrocytes (green) and 4′,6-diamidino-2-phenylindole (DAPI) was used to stain cell nuclei (blue). Scale bars, 20 µm.

**Figure 7 cells-12-01878-f007:**
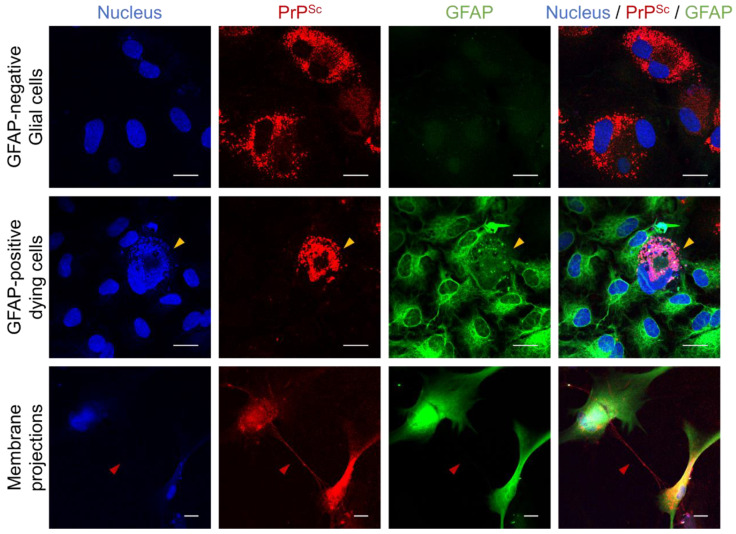
Immunocytochemistry and morphological analysis of HY-infected primary glial cultures. Primary glial cultures infected with HY prions were subjected to immunocytochemistry using mAb132 anti-PrP monoclonal antibody to detect the presence of PrP^Sc^, as described in Figure 6. The results revealed a heterogeneous population of glial cells with varying levels of GFAP expression. Not all glial cells were GFAP-positive, and the level of GFAP expression did not correlate with the susceptibility of the cells to HY infection (yellow arrowheads). Interestingly, fine membrane projections called tunneling nanotubes were observed protruding from one cell and connecting with other cells (red arrowheads). Scale bars, 20 µm.

**Figure 8 cells-12-01878-f008:**
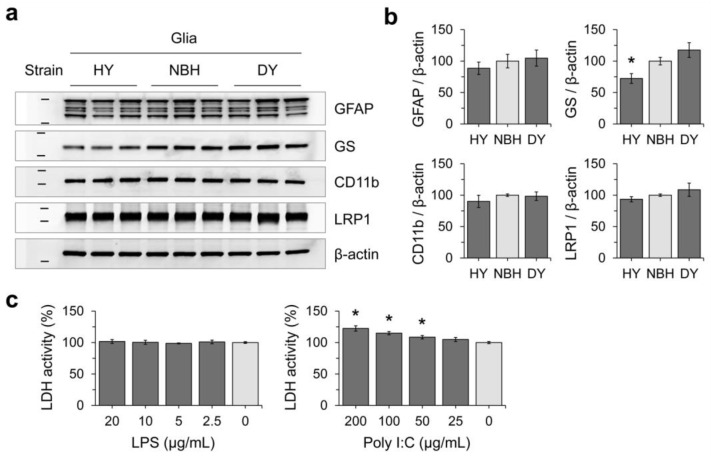
Effects of prion propagation on glial marker expression. (**a**) Western blot analysis was performed to investigate the expression of glial markers (GFAP, GS, CD11b, and LRP1) in primary glial cultures exposed to different prion strains (HY and DY) for 4 weeks. β-actin was used as a loading control. (**b**) The intensity of western blot results from (**a**) was measured and normalized to β-actin loading controls. The data were represented as mean ± SD. * *p* < 0.05 in comparison with NBH exposure controls. (**c**) The viability of primary glial cultures was assessed by measuring lactate dehydrogenase (LDH) releases after exposure to varying concentrations of lipopolysaccharides (LPS) and polyinosinic:polycytidylic acid (poly I:C). The LDH activities were expressed as a percentage of untreated controls and presented as mean ± SD. * *p* < 0.05 in comparison with the mock treatment. GFAP and GS, as astrocyte markers; CD11b as a microglial marker; and LRP1, low-density lipoprotein receptor-related protein 1. Non-infectious brain homogenates (NBHs) were used as negative controls.

**Figure 9 cells-12-01878-f009:**
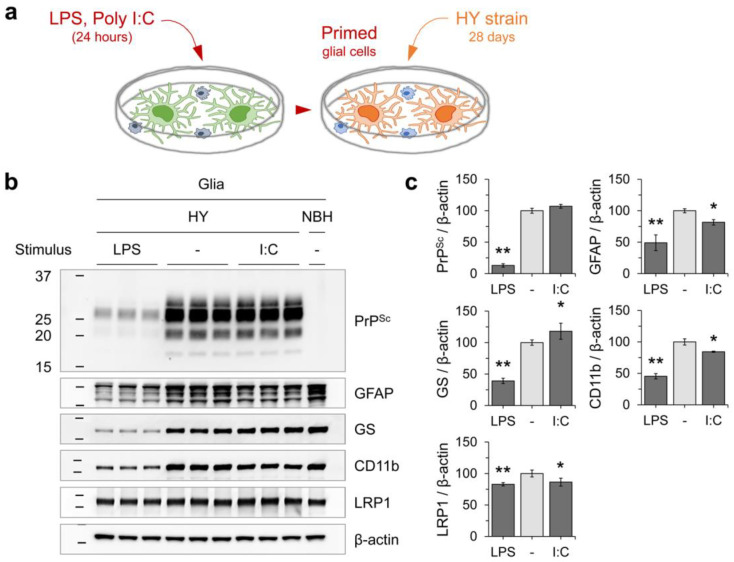
Decrease in HY prion propagation after stimulating innate immune responses. (**a**,**b**) Primary glial cultures were pre-treated with either LPS (10 μg/mL) or poly I:C (100 μg/mL) for 24 h and subsequently exposed to HY prion for 4 weeks. PrP^Sc^ accumulation was measured using western blot analysis with 3F4 anti-PrP monoclonal antibody after PK digestion. The expression of glial markers (GFAP, GS, CD11b, and LRP1) were also examined, as described in Figure 8. Non-infectious brain homogenate (NBH) was used as a negative control. (**c**) The western blot results from (**b**) were quantified by measuring the intensity and normalized to β-actin loading controls. Error bars indicate the SD. Statistical significance was set at *p* < 0.05 (*) and *p* < 0.01 (**) compared to non-stimulated controls.

**Figure 10 cells-12-01878-f010:**
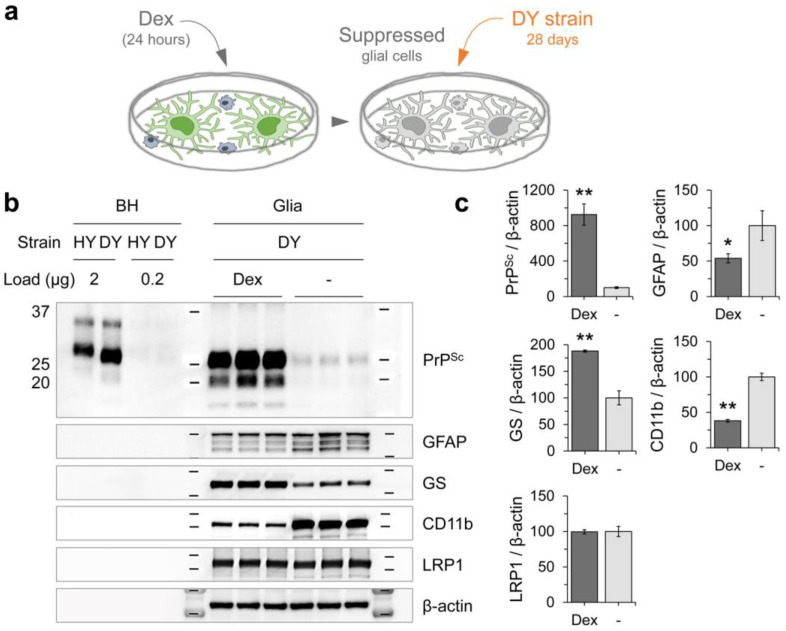
Increase in DY prion propagation after suppressing innate immune responses. (**a**,**b**) Primary glial cultures were pre-treated with dexamethasone (Dex, at 100 nM) for 24 h and subsequently exposed to DY prions for 4 weeks. PrP^Sc^ accumulation was measured using western blot analysis with 3F4 anti-PrP monoclonal antibody after PK digestion. The expression of glial markers (GFAP, GS, CD11b, and LRP1) were also examined, as described in Figure 8. HY and DY brain homogenates were included as references. (**c**) The western blot results from (**b**) were quantified by measuring the intensity and normalized to β-actin loading controls. Error bars indicate the SD. Statistical significance was set at *p* < 0.05 (*) and *p* < 0.01 (**) compared to non-suppressed controls.

**Figure 11 cells-12-01878-f011:**
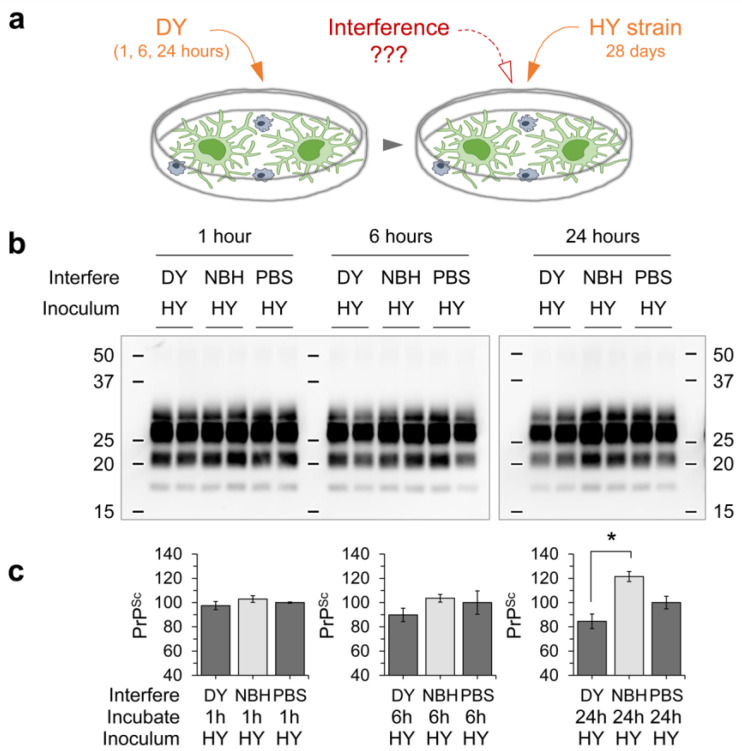
The effect of pre-exposure to DY on the propagation of HY in primary glial cultures. (**a**) The cultures were pre-exposed to DY prion (20 µg/mL) for varying time periods (1, 6, and 24 h), followed by infection with HY prion (2 µg/mL). The cells were harvested at 4 weeks post exposure to detect the presence of PrP^Sc^. Non-infectious brain homogenate (NBH) and PBS (vehicle) were used as negative controls. (**b**) PrP^Sc^ accumulation in the infected glial cells was analyzed by western blotting with 3F4 anti-PrP monoclonal antibody after PK digestion. (**c**) The western blot results from (**b**) were quantified by measuring the intensity. The error bars indicate the SD. * *p* < 0.05 in comparison with the NBH treatment.

**Figure 12 cells-12-01878-f012:**
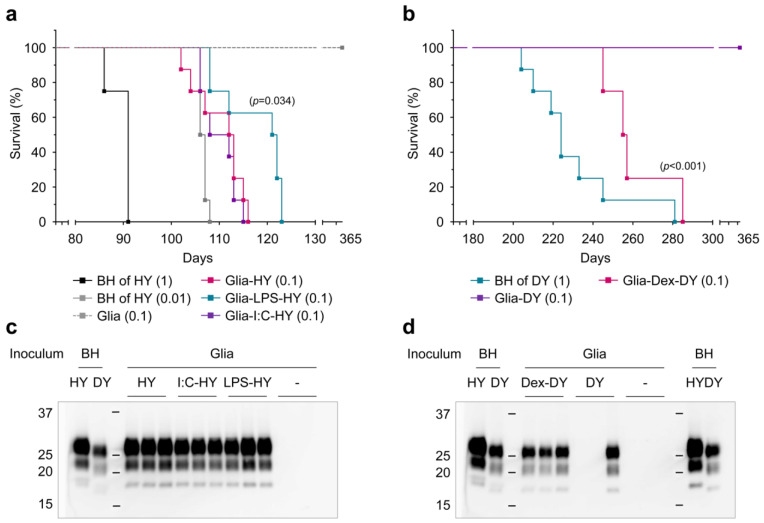
Animal bioassays examining the infectivity of primary glial cultures exposed to HY and DY prions. (**a**,**b**) Survival curves of male Syrian golden hamsters following intracerebral inoculation with varying concentrations of HY brain homogenates and HY-infected glial cells (**a**), and DY brain homogenates and DY-infected glial cells (**b**), respectively. The protein concentration (mg/mL) of each inoculum is indicated in brackets. *p*-values are included for comparisons with Glia-HY (**a**) and Glia-DY controls (**b**). (**c**,**d**) Western blot analysis of hamster brains at the clinical stage of disease from the animal bioassay in (**a**) and (**b**), respectively. PrP^Sc^ was detected by western blot using 3F4 anti-PrP monoclonal antibody after PK digestion. HY and DY brain homogenates were included as references.

**Figure 13 cells-12-01878-f013:**
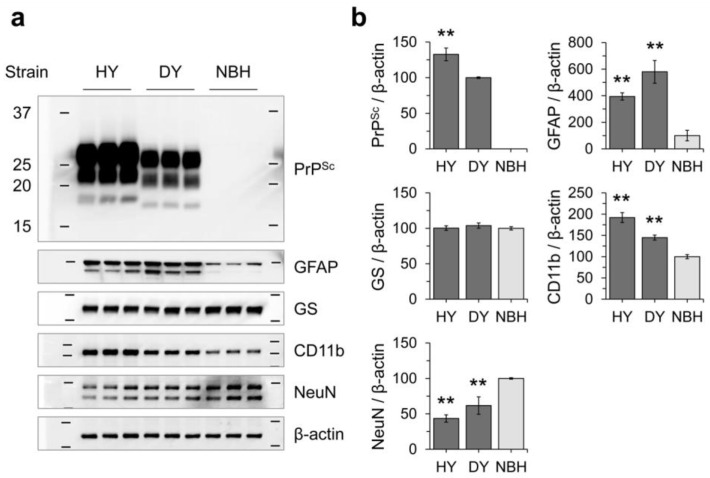
Analysis of glial marker expression in brains affected by HY and DY prions. (**a**) Western blot analysis was performed to detect PrP^Sc^ accumulation, as well as glial marker expression (GFAP, GS, CD11b, and LRP1), in the brains of hamsters that were terminally ill with HY or DY prion, respectively. The neuronal marker NeuN and β-actin loading control were also included. Non-infectious brain homogenate (NBH) was used as a negative control. (**b**) The intensity of the western blot results in (**a**) was measured and normalized to β-actin. Error bars represent the SD. Statistical analysis was performed using NBH controls as reference. Significance levels are indicated as ** *p* < 0.01.

**Figure 14 cells-12-01878-f014:**
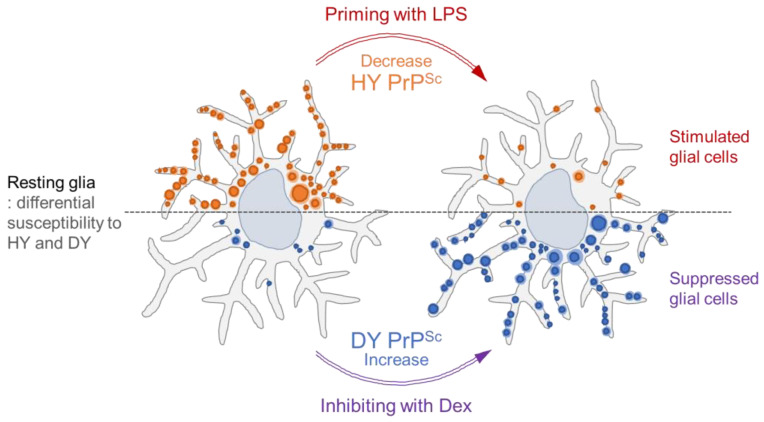
Graphic summary of glial innate immunity determining prion replication. Resting glia cells derived from the cerebellum in cultures have differential susceptibility to HY and DY prion strains, with high susceptibility to HY (oranges punctate dots) and resistance to DY (blue punctate dots). The accumulation of HY PrP^Sc^ and DY PrP^Sc^ could be influenced by innate immune modulation. When glial cells were stimulated with LPS priming, HY PrP^Sc^ significantly decreased. Conversely, when the cells were suppressed by Dex, DY PrP^Sc^ dramatically increased. It indicates that glial innate immunity plays a crucial role in determining prion replication particularly in early stage of infection.

## Data Availability

The datasets used and/or analyzed during the current study are available from the corresponding authors on reasonable request.

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
