# Peer review of "Innate Immune Status of Glia Modulates Prion Propagation in Early Stage of Infection"

_cells, 2023, doi:10.3390/cells12141878_

Round 1
Reviewer 1 Report
In this study, Kang et al highlighted the role of glial cells in the propagation of different prion strains and how the innate immune status of glial cells modulates the prion propagation in vitro and in vivo models. This is a well-conducted study. However, I have some minor concerns and suggestions-
Minor comments-
1- Authors could combine some figures together easily for better flow to read and understand.
2- Most of the western blots are overexposed authors should show low-exposure blots for better visualization and quantitation purpose. For example Figure-3a, 12 C, etc. it's hard to see any difference.
3. I understand the in vivo model is very long and time taking but it would be interesting to examine the effect of a sub-lethal dose of LPS to animals during inoculations of HY stain. Pre-incubation of glial cells to LPS has only minimal effect in animal bioassays. I'm wondering if one could modulate immune responses by treating the animals with LPS/ DEX during or after inoculation of various prion strains and examine their propagation and disease progression.
4- What is the effect of DEX treatment on glial cells with respect to HY strain propagation?
Author Response
Reviewer 1
Comments
In this study, Kang et al highlighted the role of glial cells in the propagation of different prion strains and how the innate immune status of glial cells modulates the prion propagation in vitro and in vivo models. This is a well-conducted study. However, I have some minor concerns and suggestions.
Response to Reviewer 1 comments
We sincerely appreciate your positive feedback on our manuscript. Thank you for acknowledging the significance of our research and recognizing it as a well-conducted study.
Minor comments
- Authors could combine some figures together easily for better flow to read and understand.
Response 1
We are of the opinion that preserving the original presentation will ensure the clarity and integrity of the data. We believe that making substantial alterations to the format could potentially impede the comprehension of the results. After careful consideration, we have concluded that it is best to retain the figures and western blots in their original form throughout the revised version of the manuscript.
- Most of the western blots are over exposed authors should show low-exposure blots for better visualization and quantitation purpose. For example, Figure-3a, 12 C, etc. it's hard to see any difference.
Response 2
Understood. We acknowledge your comment. In response, we have made the necessary adjustments by replacing most of the western blot results with results obtained through shorter exposure including Figures 12c and 12d.
- I understand the in vivo model is very long and time taking but it would be interesting to examine the effect of a sub-lethal dose of LPS to animals during inoculations of HY stain. Pre-incubation of glial cells to LPS has only minimal effect in animal bioassays. I'm wondering if one could modulate immune responses by treating the animals with LPS/DEX during or after inoculation of various prion strains and examine their propagation and disease progression.
Response 3
Thank you for your suggestion regarding the future directions of our research. We completely agree with your recommendation to explore the modulation of disease progression in the hamster animal model by manipulating innate immune activation/suppression. This represents the logical next step in our research, as we aim to uncover the intricate relationship between innate immunity and the pathogenesis of prion diseases.
We appreciate your observation regarding the importance of determining the appropriate dosage of LPS/Dex and carefully evaluating the response of animals, specifically the innate immune response in the brain. Our strategy involves initially identifying the specific signaling pathways as effective targets that influence prion replication in glial cells. Subsequently, we plan to translate these findings into relevant animal models to provide further insights.
We are delighted to inform you that the project is already in progress, and we are working towards achieving our research goals.
- What is the effect of DEX treatment on glial cells with respect to HY strain propagation?
Response 4
Thank you for raising this question. In theory, we expect that HY prions would propagate faster with Dex compared to the control, similar to what we observed with DY and Dex. However, it is important to consider that the presence of a high amount of HY prions may overwhelm the capacity of innate immunity to control prion propagation. This could potentially mask any preventing or clearing effect of immune modulation. Therefore, it is crucial to have well-controlled experiments that carefully consider both the input prion levels and the strength of immune modulators in order to obtain the most accurate and reliable results. By optimizing these parameters, we can ensure a comprehensive assessment of the effects of immune modulation on prion propagation, both in vitro and in vivo.
Thank you once again for your insightful suggestion and your continued interest in our work.
Best regards,
Debbie McKenzie.
Reviewer 2 Report
In this study, the authors have investigated the impact of innate immune responses on prion replication using in vitro primary glia cell culture model. The results showed PrPSc accumulation was increased with pre-treatment of dexamethasone and decreased with lipopolysaccharide. The authors suggest that neuroinflammation after prion infection is a reponse to prevent prion propagation in the brain.
The data are clearly presented and the interpretation is reasonable.
There are some minor points that the authors may consider:
1) In Fig 1 (a), the shape of cells and scale bar are unclear.
2) There are too many Figures and Fig 1 can be moved to the Supplementary.
3) Figures 2 and 3 can be combined.
4) Fig 3 (b) can be deleted.
5) Fig 4 can be moved to the Supplementary.
6) Figures 6 and 7 can be combined.
7) Figures 8-11 can be combined with the deletion of intensity data of western blot and (a) of figures 9-11.
8) Animal Bioassay and molecular genetic studies were performed with various scrapie strains including CWD-Elk. In the M & M section, the authors should explain where these studies were conducted under Biosafety Level-2 (BL2) or Biosafety Level-3 (BL3).
Author Response
Reviewer 2
Comments
In this study, the authors have investigated the impact of innate immune responses on prion replication using in vitro primary glia cell culture model. The results showed PrPSc accumulation was increased with pre-treatment of dexamethasone and decreased with lipopolysaccharide. The authors suggest that neuroinflammation after prion infection is a reponse to prevent prion propagation in the brain. The data are clearly presented and the interpretation is reasonable.
Response to Reviewer 2 comments
Thank you for your kind and generous comments on our manuscript. We appreciate your positive feedback regarding the clarity of the data presentation and the reasonableness of our interpretations.
Minor comments
- In Fig 1 (a), the shape of cells and scale bar are unclear.
Response 1
Thank you for your suggestion regarding Figure 1a. The unclear microscopic image of the primary glial cells in Figure 1a has been replaced with a clearer image. Additionally, we have highlighted the scale bar for better visibility.
- There are too many Figures and Fig 1 can be moved to the Supplementary.
- Figures 2 and 3 can be combined.
- Fig 3 (b) can be deleted.
- Fig 4 can be moved to the Supplementary.
- Figures 6 and 7 can be combined.
- Figures 8-11 can be combined with the deletion of intensity data of western blot and (a) of figures 9-11.
Response 2-7
Considering the minor comments 2 to 7, we place importance on maintaining the clarity and integrity of the data. Therefore, we are firmly committed to preserving the original presentation of the figures and western blot results. As a result, we have decided to keep the figures and western blots unchanged throughout the revision. Furthermore, we fully recognize the importance of quantification in ensuring an evaluation of the experimental results and substantiating our findings. Consequently, we have retained the quantification of western blot intensities in Figures 8 to 11 in the revised version of the manuscript, as it plays a significant role in enhancing the scientific robustness of our study.
- Animal Bioassay and molecular genetic studies were performed with various scrapie strains including CWD-Elk. In the M & M section, the authors should explain where these studies were conducted under Biosafety Level-2 (BL2) or Biosafety Level-3 (BL3).
Response 8
We fully agree with the importance of including essential information in the Materials and Methods section. Considering this, we have carefully revised the section to incorporate the biosafety level of our facility (line 176-177 in the revision) and the ethical approval number (line 179) and date (line 559).
We sincerely thank you for taking the time to review our manuscript and for providing such encouraging and constructive feedback.
Best regards,
Debbie McKenzie.